# Variational Model Inversion Attacks

**Kuan-Chieh Wang**[1,2,†]     **Yan Fu**[1]     **Ke Li**[3]     **Ashish Khisti**[1]
**Richard Zemel**[1,2]     **Alireza Makhzani**[1,2,§]
University of Toronto[1], Vector Institute[2], Simon Fraser University[3]
[†]wangkua1@cs.toronto.edu,   [§]makhzani@vectorinstitute.ai

## Abstract

Given the ubiquity of deep neural networks, it is important that these models do not reveal information about sensitive data that they have been trained on. In model inversion attacks, a malicious user attempts to recover the private dataset used to train a supervised neural network. A successful model inversion attack should generate realistic and diverse samples that accurately describe each of the classes in the private dataset. In this work, we provide a probabilistic interpretation of model inversion attacks, and formulate a variational objective that accounts for both diversity and accuracy. In order to optimize this variational objective, we choose a variational family defined in the code space of a deep generative model, trained on a public auxiliary dataset that shares some structural similarity with the target dataset. Empirically, our method substantially improves performance in terms of target attack accuracy, sample realism, and diversity on datasets of faces and chest X-ray images.

## 1   Introduction

Thanks to recent advances, deep neural networks are now widely used in applications including facial recognition [Parkhi et al., 2015], intelligent automated assistants, and personalized medicine [Consortium, 2009; Sconce et al., 2005]. Powerful deep neural networks are trained on large datasets that could contain sensitive information about individuals. Publishers release only the trained models (e.g., on TensorFlow Hub [TensorFlow]), but not the original dataset to protect the privacy of individuals whose information is in the training set. Regrettably, the published model can leak information about the original training set [Geiping et al., 2020; Nasr et al., 2019; Shokri et al., 2017]. Attacks that exploit such properties of machine learning models are privacy attacks [Rigaki and Garcia, 2020]. Other real-world scenarios where an attacker can access models trained on private datasets include interaction with APIs (e.g., Amazon Rekognition [AWS]), or in the growing paradigm of federated learning [Bonawitz et al., 2019; Yang et al., 2019a], where a malicious insider can try to steal data from other data centers [Hitaj et al., 2017]. Kaissis et al. [2020] provides an overview of how these privacy concerns are hindering the widespread application of deep learning in the medical domain.

This paper focuses on the model inversion (MI) attack, a type of privacy attack that tries to recover the training set given access only to a trained classifier [Chen et al., 2020; Fredrikson et al., 2015, 2014; Hidano et al., 2017; Khosravy et al., 2020; Yang et al., 2019b; Zhang et al., 2020]. The classifier under attack is referred to as the "target classifier" (see Figure 1). An important relevant application concerns identity recognition from face images, since facial recognition is a common approach to biometric identification [Galbally et al., 2014; Rathgeb and Uhl, 2011]. Once an attacker successfully reconstructs images of the faces of individuals in the private training set, they can use the stolen identity to break into otherwise secure systems.

Early studies of MI attacks focused on attacking classifiers whose inputs are tabular data [Fredrikson et al., 2015, 2014; Hidano et al., 2017] by directly performing inference in the input space. However, the same methods fail when the target classifier processes high-dimensional inputs, e.g., in the above

35th Conference on Neural Information Processing Systems (NeurIPS 2021), held virtually.

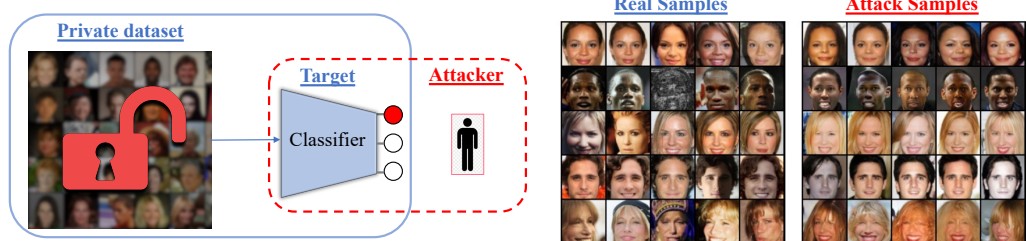

Figure 1: In model inversion attacks, the attacker tries to recover the original dataset used to train a classifier with access to only the trained classifier. In the two figures on the right, each row corresponds to a private identity – an individual whose images are not available to the attacker. The first five identities (not cherry-picked) were visualized. On the left are real data samples, and on the right are generated samples from our proposed method.

example of a face recognizer. Recent methods cope with this high-dimensional search space by performing the attack in the latent space of a neural network generator [Khosravy et al., 2020; Yang et al., 2019b; Zhang et al., 2020]. Introducing a generator works well in practice, but this success is not theoretically understood. In this paper, we frame MI attacks as a variational inference (VI) [Blei et al., 2017; Jordan et al., 1999; Kingma and Welling, 2013] problem, and derive a practical objective. This view allows us to justify existing approaches based on a deep generative model, and identify missing ingredients in them. Empirically, we find that our proposed objective leads to significant improvements: more accurate and diverse attack images, which are quantified using standard sample quality metrics (i.e., FID [Heusel et al., 2017], precision, and recall [Kynkäänniemi et al., 2019; Naeem et al., 2020]).

In summary, our contributions and findings are:

- We view the model inversion (MI) attack problem as a variational inference (VI) problem. This view allows us to derive a practical objective based on a statistical divergence, and provides a unified framework for analyzing existing methods.

- We provide an implementation of our framework using a set of deep normalizing flows [Dinh et al., 2014, 2016; Kingma and Dhariwal, 2018] in the extended latent space of a Style-GAN [Karras et al., 2020]. This implementation can leverage the hierarchy of learned representations, and perform targeted attacks while preserving diversity.

- Empirically, on the CelebA [Liu et al., 2015], and ChestX-ray [Wang et al., 2017] datasets, our method results in higher attack accuracy, and more diverse generation compared to existing methods. We provide thorough comparative experiments to understand which components contribute to improved target accuracy, sample realism, and sample diversity, and detail the benefits of the VI perspective.

Code can be found at `https://github.com/wangkua1/vmi`.

## 2 Problem Setup: Model Inversion Attack

In the problem of model inversion (MI) attack, the attacker has access to a "target classifier":

$$\overline{p}_{\text{TAR}}(y|\mathbf{x}) : \mathbb{R}^d \to \Delta^{C-1}$$

where $\Delta^{C-1}$ to denote the $(C-1)$-simplex, representing the $C$-dimensional probability where $C$ is the number of classes. This target classifier is trained on the private target dataset, $\mathcal{D}_{\text{TAR}} = \{\mathbf{x}_i, y_i\}_{i=1}^{N_{\text{TAR}}}$ where $\mathbf{x} \in \mathbb{R}^d$ and $y \in \{1, 2, ..., C\}$. We use $\overline{p}_{\text{TAR}}(y|\mathbf{x})$ to denote the given target classifier, which is an approximation of the true conditional probability $p_{\text{TAR}}(y|\mathbf{x})$ of the underlying data generating distribution.

**Goal.** Given a target classifier $\overline{p}_{\text{TAR}}(y|\mathbf{x})$, we wish to approximate the class conditional distribution $p_{\text{TAR}}(\mathbf{x}|y)$ without having access to the private training set $\mathcal{D}_{\text{TAR}}$.

A good model inversion attack should approximate the Bayes posterior $p_{\text{TAR}}(\mathbf{x}|y) \propto p_{\text{TAR}}(y|\mathbf{x})p_{\text{TAR}}(\mathbf{x})$ well, and allow the attacker to generate realistic, accurate, and diverse samples. Notice, our goal shifts slightly from recovering the exact instances in the training set to recovering the data generating distribution of the training data. Let's consider a scenario where the attacker intends to fake a victim's identity. A good security system might test the realism of a sequence of inputs [Holland and Komogortsev, 2013]. More generally, faking someone's identity requires passing a two-sample test [Holland and Komogortsev, 2013; Lehmann and Romano, 2006], which means the ability to generate samples from $p_{\text{TAR}}(\mathbf{x}|y)$ is necessary. Another scenario can be an attacker trying to steal private/proprietary information. For example, a malicious insider in a federated learning setting can try to steal information from another datacenter by accessing the shared model [Hitaj et al., 2017]. In such a scenario, a good $p_{\text{TAR}}(\mathbf{x}|y)$ model can reveal not only the distinguishing features of the data, but also its variations.

## 3  Related Work

In this section, we first discuss related model inversion attack studies, and then discuss applications of inverting a classifier outside of the privacy attack setting. Lastly, since our attack method relies on using a pretrained generator, we discuss existing studies that used a pretrained generator for improving other applications.

**Model inversion attacks.**  MI attacks are one type of privacy attack where the attacker tries to reconstruct the training examples given access only to a target classifier. Other privacy attacks include membership attacks, attribute inference, and model extraction attacks (see Rigaki and Garcia [2020] for a general discussion). Fredrikson et al. [Fredrikson et al., 2015, 2014] were the first to study the MI attack and demonstrated successful attacks on low-capacity models (e.g., logistic regression, and a shallow MLP network) when partial input information was available to the attacker. The setting where partial input information was not available was studied by Hidano et al. [2017]. They termed performing gradient ascent w.r.t. the target classifier in the input space a *General Model Inversion* attack. Though effective on tabular data, this approach failed on deep image classifiers. It is well-known that directly performing gradient ascent in the image space results in imperceptible and unnatural changes [Szegedy et al., 2013].

To handle high-dimensional observations such as images, MI attacks based on deep generators were proposed [Chen et al., 2020; Yang et al., 2019b; Zhang et al., 2020]. By training a deep generator on an *auxiliary dataset*, these methods effectively reduce the search space to only the manifold of relevant images. For example, if the target classifier is a facial recognition system, then an auxiliary dataset of generic faces can be easily obtained. Zhang et al. [2020] pretrain a Generative Adversarial Network (GAN) [Goodfellow et al., 2014] on the auxiliary dataset and performed gradient-based attack in the latent space of the generator. Yang et al. [2019b] trained a generator that inverts the prediction of the target classifier, using the architecture of an autoencoder.

**Other applications of classifier inversion.**  Inverting the predictions and activations of a classifier has been studied for applications such as model explanation, and model distillation. Motivated by using saliency maps to explain the decision of a classifier [Simonyan et al., 2013], Mahendran and Vedaldi [2015] proposed to maximize the activation of a target neuron in a classifier for synthesizing images that best explain that neuron. Their method is based on gradient ascent in the pixel-space. The same idea was recently extended to enable knowledge distillation in a setting where the original dataset was too large to be transferred [Yin et al., 2020]. Methods above have the advantage of not requiring the original dataset, but the generated images still appear unnatural. One effective way to improve the realism of the generated images is to optimize in the latent code space of a deep generator [Nguyen et al., 2015, 2017].

**Pretrained generators.**  Pretrained generators from GANs have other applications. Image inpainting/restoration was one of the early successful applications of using a pretrained GAN as image prior. Yeh et al. [2017] optimized the latent code of GANs to minimize the reconstruction error of a partially occluded image for realistic inpainting. The same method formed the basis of AnoGAN, a popular technique for anomaly detection in medical images [Schlegl et al., 2017]. A combination of reconstruction error and discriminator loss was used as the anomaly score. A potential future direction is to apply our proposed method to these adjacent applications.

## 4 Model Inversion Attack as Variational Inference

In this section, we derive our *Variational Model Inversion (VMI)* learning objective. For each class label $y$, we wish to approximate the true target posterior with a variational distribution $q(\mathbf{x}) \in \mathcal{Q}_\mathbf{x}$, where $\mathcal{Q}_\mathbf{x}$ is the variational family. This can be achieved by the optimizing the following objectives

$$
\begin{aligned}
q^*(\mathbf{x}) &= \arg\min_{q \in \mathcal{Q}_\mathbf{x}} \left\{ D_{\mathrm{KL}}(q(\mathbf{x}) || p_{\mathrm{TAR}}(\mathbf{x}|y)) \right\} \\
&= \arg\min_{q \in \mathcal{Q}_\mathbf{x}} \left\{ \mathbb{E}_{q(\mathbf{x})}[-\log p_{\mathrm{TAR}}(y|\mathbf{x})] + D_{\mathrm{KL}}(q(\mathbf{x}) || p_{\mathrm{TAR}}(\mathbf{x})) + \log p_{\mathrm{TAR}}(y) \right\} \\
&= \arg\min_{q \in \mathcal{Q}_\mathbf{x}} \left\{ \mathbb{E}_{q(\mathbf{x})}[-\log p_{\mathrm{TAR}}(y|\mathbf{x})] + D_{\mathrm{KL}}(q(\mathbf{x}) || p_{\mathrm{TAR}}(\mathbf{x})) \right\}.
\end{aligned}
\tag{1}
$$

where in Equation 1 we have used the fact that $\log p_{\mathrm{TAR}}(y)$ is independent of $q(\mathbf{x})$, and thus can be dropped in the optimization.

**Auxiliary Image Prior.** In order to minimize Eq. (1), we need to compute $D_{\mathrm{KL}}(q(\mathbf{x})||p_{\mathrm{TAR}}(\mathbf{x}))$, which requires evaluating the true image prior $p_{\mathrm{TAR}}(\mathbf{x})$, that is not accessible. Similar to recent works on MI attacks [Khosravy et al., 2020; Yang et al., 2019b; Zhang et al., 2020], we assume that a public auxiliary dataset $\mathcal{D}_{\mathrm{AUX}}$, which shares structural similarity with the target dataset, is available to learn an auxiliary image prior $p_{\mathrm{AUX}}(\mathbf{x})$. This auxiliary image prior $p_{\mathrm{AUX}}(\mathbf{x})$ is learned using a generative adversarial network (GAN) $p_{\mathrm{AUX}}(\mathbf{x}) = \mathbb{E}_{p_{\mathrm{AUX}}(\mathbf{z})}[p_{\mathrm{G}}(\mathbf{x}|\mathbf{z})]$, where $p_{\mathrm{AUX}}(\mathbf{z}) = \mathcal{N}(0, I)$ is the GAN prior, and $p_{\mathrm{G}}(\mathbf{x}|\mathbf{z})$ is the GAN generator that is parameterized by a deep neural network $G(\mathbf{z})$, and a Gaussian observation noise with small standard deviation $\sigma$: $p_{\mathrm{G}}(\mathbf{x}|\mathbf{z}) = \mathcal{N}(G(\mathbf{z}), \sigma^2 I)$, or equivalently

$$
\mathbf{x} = G(\mathbf{z}) + \sigma\boldsymbol{\epsilon}, \quad \boldsymbol{\epsilon} \sim \mathcal{N}(0, I).
$$

The learned generator $p_{\mathrm{G}}(\mathbf{x}|\mathbf{z})$ defines a low dimensional manifold in the observation space $\mathbf{x}$, when $\mathbf{z}$ is chosen to have fewer dimensions than $\mathbf{x}$. We now make the following assumption about the structural similarity of $p_{\mathrm{AUX}}(\mathbf{x})$ and $p_{\mathrm{TAR}}(\mathbf{x})$.

**Assumption 1** (Common Generator). *We assume $p_{\mathrm{TAR}}(\mathbf{x})$ is concentrated close to the low dimensional manifold of $p_{\mathrm{G}}(\mathbf{x}|\mathbf{z})$, learned from the auxiliary dataset $\mathcal{D}_{\mathrm{AUX}}$. In other words, we assume there exists a target prior $p_{\mathrm{TAR}}(\mathbf{z})$, possibly different from $p_{\mathrm{AUX}}(\mathbf{z})$, such that $p_{\mathrm{TAR}}(\mathbf{x}) \approx \mathbb{E}_{p_{\mathrm{TAR}}(\mathbf{z})}[p_{\mathrm{G}}(\mathbf{x}|\mathbf{z})]$. We refer to $p_{\mathrm{G}}(\mathbf{x}|\mathbf{z})$ as the "common generator".*

The name "common generator" is inspired by recent progress in theoretical analyses of few-shot and transfer learning, where assumptions about an underlying shared representation have been used to provide error bounds. In these works, this shared representation is referred to as the "common representation" [Du et al., 2020; Tripuraneni et al., 2020]. Assumption 1 usually holds in the context of model inversion attacks. For example, $p_{\mathrm{AUX}}(\mathbf{x})$ could be the distribution of all human faces, while $p_{\mathrm{TAR}}(\mathbf{x})$ could be the distribution of the celebrity faces; or $p_{\mathrm{AUX}}(\mathbf{x})$ could be the distribution of natural images, while $p_{\mathrm{TAR}}(\mathbf{x})$ could be the distribution of different breeds of dogs. In these examples, both $p_{\mathrm{AUX}}(\mathbf{x})$ and $p_{\mathrm{TAR}}(\mathbf{x})$ approximately live close to the same low dimensional manifold, but the distribution on this shared manifold could be different, i.e., $p_{\mathrm{TAR}}(\mathbf{z}) \neq p_{\mathrm{AUX}}(\mathbf{z})$.

**Variational Family.** We further consider our variational family $\mathcal{Q}_\mathbf{x}$ to be all distributions that lie on this manifold by assuming that $q(\mathbf{x})$ is of the form $q(\mathbf{x}) = \mathbb{E}_{q(\mathbf{z})}[p_{\mathrm{G}}(\mathbf{x}|\mathbf{z})]$, $q(\mathbf{z}) \in \mathcal{Q}_\mathbf{z}$, where $\mathcal{Q}_\mathbf{z}$ is a variational family in the code space. In the next section, we further restrict $\mathcal{Q}_\mathbf{z}$ to be either Gaussian distributions or normalizing flows.

**Proposition 1.** *We have $D_{\mathrm{KL}}(q(\mathbf{z})||p_{\mathrm{TAR}}(\mathbf{z})) \geq D_{\mathrm{KL}}(\mathbb{E}_{q(\mathbf{z})}[p_{\mathrm{G}}(\mathbf{x}|\mathbf{z})]||\mathbb{E}_{p_{\mathrm{TAR}}(\mathbf{z})}[p_{\mathrm{G}}(\mathbf{x}|\mathbf{z})])$.*

See Appendix A for the proof. Using Assumption 1, we can approximate the objective of Equation 1 with

$$
q^*(\mathbf{z}) = \arg\min_{q \in \mathcal{Q}_\mathbf{z}} \left\{ \mathbb{E}_{\mathbf{z} \sim q(\mathbf{z}), \boldsymbol{\epsilon} \sim \mathcal{N}(0,I)}[-\log p_{\mathrm{TAR}}(y|G(\mathbf{z}) + \sigma\boldsymbol{\epsilon})] + D_{\mathrm{KL}}(\mathbb{E}_{q(\mathbf{z})}[p_{\mathrm{G}}(\mathbf{x}|\mathbf{z})]||\mathbb{E}_{p_{\mathrm{TAR}}(\mathbf{z})}[p_{\mathrm{G}}(\mathbf{x}|\mathbf{z})]) \right\},
\tag{2}
$$

where this objective is now defined in terms of $q(\mathbf{z}) \in \mathcal{Q}_\mathbf{z}$ in the latent space of the GAN. Now using the Proposition 1, we can obtain the following upper bound on the objective of Equation 2 (assuming $\sigma = 0$)

$$
q^*(\mathbf{z}) = \arg\min_{q \in \mathcal{Q}_\mathbf{z}} \left\{ \mathbb{E}_{\mathbf{z} \sim q(\mathbf{z})}[-\log p_{\mathrm{TAR}}(y|G(\mathbf{z}))] + D_{\mathrm{KL}}(q(\mathbf{z})||p_{\mathrm{TAR}}(\mathbf{z})) \right\}.
\tag{3}
$$

**Power Posteriors.** We now consider the optimization of Equation 3. Since we do not have access to $p_{\text{TAR}}(\mathbf{z})$, we propose to replace it with $p_{\text{AUX}}(\mathbf{z})$, noting that, as argued above, $p_{\text{AUX}}(\mathbf{z})$ may not accurately represent $p_{\text{TAR}}(\mathbf{z})$. We also propose to replace the unknown likelihood $p_{\text{TAR}}(y|G(\mathbf{z}))$ with the $\overline{p}_{\text{TAR}}(y|G(\mathbf{z}))$ defined by the target classifier, noting that this likelihood could be miscalibrated. Miscalibration is a known problem in neural network classifiers [Guo et al., 2017]; namely, the classifier predictions can be over- or under-confident. In order to account for these issues, instead of the standard Bayes posterior $q_{\text{BAYES}}(\mathbf{z}) \propto p_{\text{AUX}}(\mathbf{z})\overline{p}_{\text{TAR}}(y|G(\mathbf{z}))$ as the solution of Equation 3, we consider the family of "power posterior" [1] [Bissiri et al., 2016; Holmes and Walker, 2017; Knoblauch et al., 2019; Miller and Dunson, 2018], which raise the likelihood function to a power $\frac{1}{\gamma}$

$$q_\gamma^*(\mathbf{z}) \propto p_{\text{AUX}}(\mathbf{z})\overline{p}_{\text{TAR}}^{\frac{1}{\gamma}}(y|G(\mathbf{z})),$$

and can be characterized as the solution of the following optimization problem:

$$q_\gamma^*(\mathbf{z}) = \underset{q \in \mathcal{Q}_{\mathbf{z}}}{\arg\min} \, \mathcal{L}_{\text{VMI}}^\gamma(q), \tag{4}$$

$$\mathcal{L}_{\text{VMI}}^\gamma(q) := \mathbb{E}_{\mathbf{z} \sim q(\mathbf{z})}[-\log \overline{p}_{\text{TAR}}(y|G(\mathbf{z}))] + \gamma D_{\text{KL}}(q(\mathbf{z})||p_{\text{AUX}}(\mathbf{z})).$$

See Appendix A for the proof. If the prior $p_{\text{AUX}}(\mathbf{z})$ is not reliable, by decreasing $\gamma$, we can reduce the importance of the prior. Similarly, by adjusting $\gamma$, we can reduce or increase the importance of over- or under-confident likelihood predictions $\overline{p}_{\text{TAR}}(y|G(\mathbf{z}))$.

**Accuracy vs. Diversity Tradeoff.** In the context of model inversion attack, the Bayes posterior $q_{\text{BAYES}}(\mathbf{z})$ can only achieve a particular tradeoff between the accuracy and diversity of the generated samples. However, the family of power posteriors enables us to achieve an arbitrary desired tradeoff between accuracy and diversity by tuning $\gamma$. If $\gamma = 0$, then $q_{\gamma=0}^*(\mathbf{z}) = \delta(\mathbf{z} - \mathbf{z}^*)$, and the model essentially ignores the image prior and diversity, and finds a single point estimate $\mathbf{z}^*$ on the manifold which maximizes the accuracy $\overline{p}_{\text{TAR}}(y|G(\mathbf{z}))$. As we increase $\gamma$, the the model achieves larger diversity at the cost of smaller average accuracy. When $\gamma = 1$, we recover the Bayes posterior $q_{\gamma=1}^*(\mathbf{z}) \propto p_{\text{AUX}}(\mathbf{z})\overline{p}_{\text{TAR}}(y|G(\mathbf{z}))$. In the limit of $\gamma \to \infty$, we have $q_\infty^*(\mathbf{z}) = p_{\text{AUX}}(\mathbf{z})$. In this case, the model ignores the classifier and accuracy, and sample unconditionally from the auxiliary distribution (maximum diversity). We empirically study the effect of $\gamma$ in Section 5.

### 4.1 The Common Generator

In the previous section, we derived a practical MI objective from the perspective of variational inference. Similar to previous works [Chen et al., 2020; Zhang et al., 2020] VMI requires a common generator that is trained on a relevant auxiliary dataset. In this subsection, we first provide a brief description of the most common choice, DCGAN. Then, we introduce StyleGAN, which has an architecture that allows for fine-grained control. Finally, we describe how to adapt our VMI objective to leverage this architecture by using a layer-wise approximate distribution.

For the common generator $G$ we use a GAN due to their ability to synthesize high quality samples [Abdal et al., 2019; Bau et al., 2018; Karras et al., 2019; Shen et al., 2020]. The generator is trained on the auxiliary dataset. A common choice is the DCGAN architecture [Radford et al., 2015]. Though optimizing in the latent space results in more realistic samples, this search space is also more restrictive. Early investigation showed that optimizing at some intermediate layers of a pretrained generator was better than either optimizing in the code space, or the pixel space. This motivated the use of a generator architecture where activations of intermediate layers can be manipulated.

**StyleGAN.** A framework that naturally allows for optimization in the intermediate layers is the StyleGAN [Karras et al., 2019]. It consists of a mapping network $f : \mathbf{z} \mapsto \mathbf{w}$, and a synthesis network $S$. The synthesis network takes as input one $\mathbf{w}$ at each layer, $S : \{\mathbf{w}_l\}_{l=1}^L \mapsto \mathbf{x}$, where $L$ is the number of layers in $S$. During training of a StyleGAN, the output from the mapping network is copied $L$ times, $\{\mathbf{w}\}_{l=1}^L = \text{COPY}(\mathbf{w}, L)$ before being fed to the synthesis network. This input space of the synthesis network $S$ has been referred to as the "extended $\mathbf{w}$ space" [Abdal et al., 2019]. The full generator is $G_{\text{STYLE}}(\mathbf{z}) = S\big(\text{COPY}(f(\mathbf{z}), L)\big)$.

---

[1] These posterior updates are sometimes referred to as "power likelihood" or "power prior" [Bissiri et al., 2016].

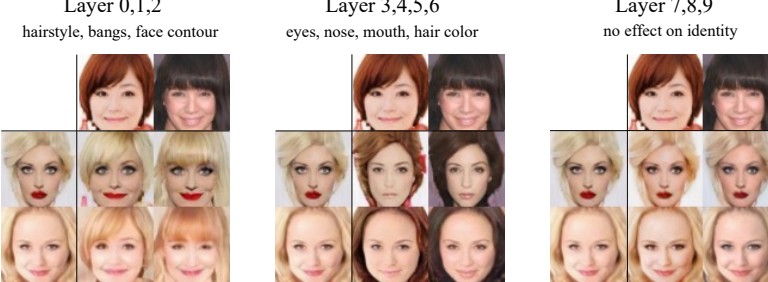

Layer 0,1,2
hairstyle, bangs, face contour

Layer 3,4,5,6
eyes, nose, mouth, hair color

Layer 7,8,9
no effect on identity

Figure 2: Visualizing the factors of variations in each layer of the StyleGAN [Karras et al., 2019] using style mixing. Each image is generated by mixing the the corresponding image in the topmost row, and the image in the leftmost column. The topmost row image provides the features for the indicated set of layers, and the leftmost column image provides features for the remaining layers. We can see that in general, the earlier features contribute more to the facial features, while the later features contribute more to the low-level images statistics such as lighting and general shading. Given this learned layer-wise disentanglement, our VMI attack can adaptively discover layers that contribute the most to the face identity (layers $1, 2, 4, 5, 6$), and only manipulates the corresponding features (see Figure 7). This is not facilitated by the generator of a DCGAN, as found in existing MI attack methods [Chen et al., 2020; Zhang et al., 2020].

StyleGAN is known for its surprising ability to perform "style mixing". Namely, the synthesis network can generate a "mixed" image, when being fed a mixture of two $\mathbf{w}$'s in the extended $\mathbf{w}$ space, i.e., $S([\{\mathbf{w}_1\}_{l=1}^{L_0}; \{\mathbf{w}_2\}_{l=L_0+1}^{L}])$, where $[\cdot; \cdot]$ denotes concatenation, and $L_0$ is some intermediate layer. Figure 2 visualizes the effect of manipulating different layers of the synthesis network. Performing an attack in this extended $\mathbf{w}$ space allows us to achieve the desired effect of optimizing the intermediate layers. Because of the lack of a explicit prior for $\mathbf{w}$, we instead optimize the extended $\mathbf{z}$ space (Figure 3). Suppose $q(\mathbf{z}_1, \ldots, \mathbf{z}_L)$ is the joint density over

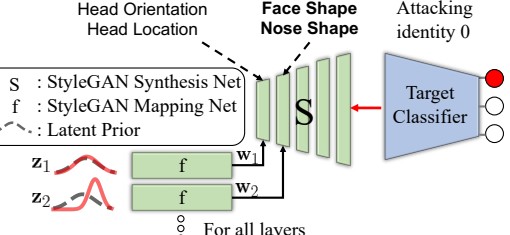

Figure 3: Utilizing the StyleGAN architecture, our VMI attack tends to focus on layers whose representation are relevant for the attack, such as the layer for "face shape".

$\mathbf{z}_1, \ldots, \mathbf{z}_L$, and $q_l(\mathbf{z}_l)$ is the marginal density over $\mathbf{z}_l$. We consider the following objective for the VMI with the StyleGAN generator:

$$\mathcal{L}_{\text{S-VMI}}^{\gamma}(q) := \mathbb{E}_{q(\mathbf{z}_1, \ldots, \mathbf{z}_L)} \Big[ - \log \bar{p}_{\text{TAR}} \Big( y \Big| S\big(\{f(\mathbf{z}_l)\}_{l=1}^{L}\big) \Big) \Big] + \frac{\gamma}{L} \sum_{l=1}^{L} D_{\text{KL}}(q_l(\mathbf{z}_l) || p_{\text{AUX}}(\mathbf{z}_l)). \quad (5)$$

### 4.2 Code Space Variational Family $\mathcal{Q}_{\mathbf{z}}$

In the VMI formulations (Equation 4 and 5), the approximate distribution $q(\mathbf{z})$ can be seen as a generator, or "miner" network [Wang et al., 2020] for the latent space of the common generator. It must belong to a distribution family $\mathcal{Q}_{\mathbf{z}}$, whose distribution can be sampled from, and can be evaluated in order to optimize the KL-divergence. In this work, we experiment with a powerful class of tractable generative models: deep flow models. When optimizing $\mathcal{L}_{\text{S-VMI}}^{\gamma}(q)$ (Equation 5), we use $L$ such models, one for each layer in the synthesis network. In our experiments, we used an architecture similar to that of Glow [Kingma and Dhariwal, 2018], where we treated the latent vector as 1x1 images. We removed the original squeezing layers that reduced the image size to account for this change. We also experimented with using a Gaussian variational family for $q(\mathbf{z})$.

### 4.3 Relationships with Existing Methods

VMI provides a unified framework that encompasses existing attack methods. Through the lens of VMI, we can obtain insights on the conceptual differences between the existing methods and their pros and cons. "General MI" attack [Hidano et al., 2017] performs the following optimization:

$$\mathbf{x}^* = \arg\max_{\mathbf{x}} \log \bar{p}_{\text{TAR}}(y|\mathbf{x}). \quad (6)$$

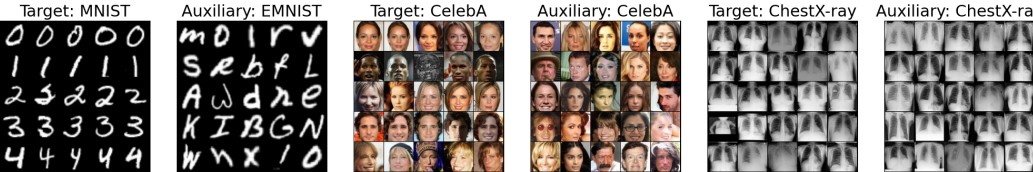

Figure 4: Visualization of real samples for all 3 tasks: MNIST, CelebA, and ChestX-ray. For the target datasets, each row corresponds to one class/identity. Best viewed zoomed in.

Note that this objective can be viewed as special case of VMI, where the code space is the same as the data space, and we have $\gamma = 0$ in the objective: $\mathcal{L}_{\text{VMI}}^{\gamma=0}(q)$. In this case, as discussed in the previous section, the $q(\mathbf{x})$ distribution collapses to a point estimate at which $\overline{p}_{\text{TAR}}(y|\mathbf{x})$ is maximized. As we will see in the experiment section, since the optimization is in the data space, the solution does not live close the manifold of images, and thus generally does not structurally looks like natural images.

"Generative MI" attack [Zhang et al., 2020] also uses a generator trained on an "auxiliary" dataset. The attack optimizes the following objective:

$$\mathbf{z}^* = \arg\min_{\mathbf{z}} \left\{ -\lambda \log \overline{p}_{\text{TAR}}(y|(G(\mathbf{z})) - \log \sigma\big(D(G(\mathbf{z}))\big) \right\} \tag{7}$$

where $G, D$ denotes the generator and discriminator respectively, and $\sigma(\cdot)$ is the sigmoid function. Note that this objective can be viewed as special case of VMI, with $\gamma = \frac{1}{\lambda}$, and with the prior $p(\mathbf{z}) \propto \sigma\big(D(G(\mathbf{z}))\big)$, and a point estimate for $\mathbf{z}$, instead of optimizing over distributions $q(\mathbf{z})$. As a result of this point-wise optimization, the Generative MI attack requires re-running the optimization procedure for every new samples. In contrast, VMI learns a distribution $q(\mathbf{z})$ using expressive normalizing flows, which allows us to draw multiple attack samples after a single optimization procedure. Recently, Chen et al. [2020] proposed extending Generative MI by considering a Gaussian variational family in the latent space of the pretrained generator in Equation 7. However, we note this method still theoretically results in collapsing the Gaussian variational distributions to a point estimate. VMI prevents this by penalizing the entropy of the variational distribution with the KL-divergence term. From the implementation perspective, Chen et al. [2020] uses DCGAN while we use StyleGAN which helps us to achieve better disentanglement. Furthermore, we use a more expressive variational family such as normalizing flows.

## 5 Experiments

In this section, we demonstrate the effectiveness of the proposed VMI attack method on three separate tasks. We will refer to each task by their target dataset: MNIST [LeCun et al., 2010], CelebA [Liu et al., 2015], and ChestX-ray (CXR) [Wang et al., 2017]. Then, we showcase the robustness of the VMI attack to different target classifiers on CelebA. More experimental details can be found in Appendix B.

**Data.** For the MNIST task, we used the 'letters' split (i.e., handwritten English alphabets) of the EMNIST [Cohen et al., 2017] dataset as the auxiliary dataset. For CelebA, we used the 1000 most frequent identities as the target dataset, and the rest as the auxiliary dataset. For ChestX-ray, we used the 8 diseases outlined by Wang et al. [2017] as the target dataset, and randomly selected 50,000 images from the remaining as the auxiliary dataset. Real samples from these six datasets are visualized in Figure 4.

**Target classifiers.** The target classifiers used in all tasks were ResNets trained using SGD with momentum. Optimization hyperparameters were selected to maximize accuracy on a validation set of the private data.

**Evaluation metrics.** An ideal evaluation protocol should measure different aspects of the attack samples such as *target accuracy*, *sample realism*, and *sample diversity*.

*Target Accuracy*: To automate the process of judging the accuracy of an attack, we use an "evaluation" classifier. This metrics measures how well the generated samples resemble the target class/identity. Given a target identity $i$, an attack produces a set of samples $\{\mathbf{x}'_n\}_{n=1}^N$. Each sample is assigned a predicted label by the evaluation classifier, and then the top-1 accuracy is computed.

|  | Accuracy↑ | | | FID↓ | | |
|---|---|---|---|---|---|---|
|  | MNIST | CelebA | CXR | MNIST | CelebA | CXR |
| **General MI** [Hidano et al., 2017] | 0 ± 0.00 | 0± 0.00 | 0.23± 0.29 | 376.7 (57.4) | 421.21 (31.3) | 499.54 (96.3) |
| **Generative MI** [Zhang et al., 2020] | 0.92 ± 0.02 | 0.07± 0.02 | 0.28± 0.25 | 88.91 (57.4) | 43.21 (31.3) | 142.66 (96.3) |
| **VMI w/ DCGAN** | **0.95**± 0.02 | 0.37± 0.07 | 0.42± 0.28 | **77.73** (57.4) | 40.89 (31.3) | 265.14 (96.3) |
| **VMI w/ StyleGAN** | - | **0.55**± 0.06 | **0.69**± 0.23 | - | **17.41** (19.2) | **123.17** (57.0) |

Table 1: MI attack comparison across tasks. Both VMI methods here used the Flow model for $q(\mathbf{z})$. Best values are in ***bold***, and values within the 95% confidence interval ($\pm$) of the best are *underlined*. The numbers in the braces in the FID columns are FIDs of the pre-trained generators. Our VMI attack results in better accuracy and FID compared to baselines on all three tasks.

If a good evaluation classifier only generalized to realistic inputs, then this metric would also penalize unrealistic attack samples. To some extent, this was the case in our experiments as evidenced by the extremely low accuracy assigned to the General MI attack baseline in Table 1. In Figure 5, the General MI attack samples were clearly unrealistic. To ensure a fair and informative evaluation, the evaluation classifier should be as accurate and generalizable as possible (details in Appendix B.3).

*Sample Realism*: We used FID [Heusel et al., 2017] to directly measure the overall sample quality. To compute FID, we aggregated attack samples across classes and compared them to aggregated real images. Following standard practice, the feature extractor used for computing FID was the pretrained Inception network [Szegedy et al., 2015].

*Sample Diversity*: Though FID captures realism and overall diversity, it is known to neglect properties such as intra-class diversity, which is an important feature of a good MI attack. To overcome this limitation, we computed the improved precision & recall [Kynkäänniemi et al., 2019], coverage & density [Naeem et al., 2020] on a per class basis, capturing the intra-class diversity of the generated samples. The same Inception network was used as the feature extractor. Recall proposed in Kynkäänniemi et al. [2019] can be problematic when the generated samples contain outliers. Coverage proposed in Naeem et al. [2020] is less susceptible to outliers, but also less sensitive. For convenience of comparison, we take the average of the two quantities and call the resulting metric "diversity".

**Generators.** Two generator models were considered: DCGAN [Radford et al., 2015], and Style-GAN [Karras et al., 2019]. For StyleGAN, we used the original implementations.[2] There were 10 and 12 layers in the synthesis networks for the 64x64 CelebA data and 128x128 ChestX-ray data respectively. We do not use StyleGAN for the MNIST task since the images are simple.

## 5.1 Results

**Comparison with Baselines.** Across all three tasks, our VMI formulations outperformed baselines in terms of target accuracy and sample realism (see Table 1), even when using the same generator architecture (DCGAN). Using a flow model for $q(\mathbf{z})$ led to improvements in terms of accuracy. Using the formulation in Equation 5 and a StyleGAN further improved performance on all metrics. VMI also improved the overall sample qualities as evidenced by the lower FIDs. The full VMI (Equation 5) was able to improve FID over pre-trained generators on CelebA. In general, MI attack methods including VMI sacrificed sample quality to achieve higher accuracy, suggesting there is still room for improvements for methods that can retain full realism. Detailed evaluation using sample diversity metrics are reported in Table 2. In general, increasing $\gamma$ increased diversity and realism while sacrificing accuracy. Our VMI attack outperformed the baseline on all metrics at multiple settings. The reported improvements in the quantitative metrics can be verified qualitatively in Figure 5. Detailed results for MNIST and ChestX-ray (Table 4, and 5) can be found in the Appendix C. Using StyleGAN with a Gaussian $q(\mathbf{z})$ resulted in a slightly lower accuracy, but better diversity.

Our reported results for our baseline [Zhang et al., 2020] are obtained using our own implementation since the reproduction code was not publicly available. The discrepancy with the original result can come from a number of factors: dataset split, target classifier checkpoint, and the hyperparameters involved in the method. We used the task setup described in this paper, and tuned the hyperparameters over a reasonable grid of values.

---

[2]Code from: `https://github.com/NVlabs/stylegan2-ada`

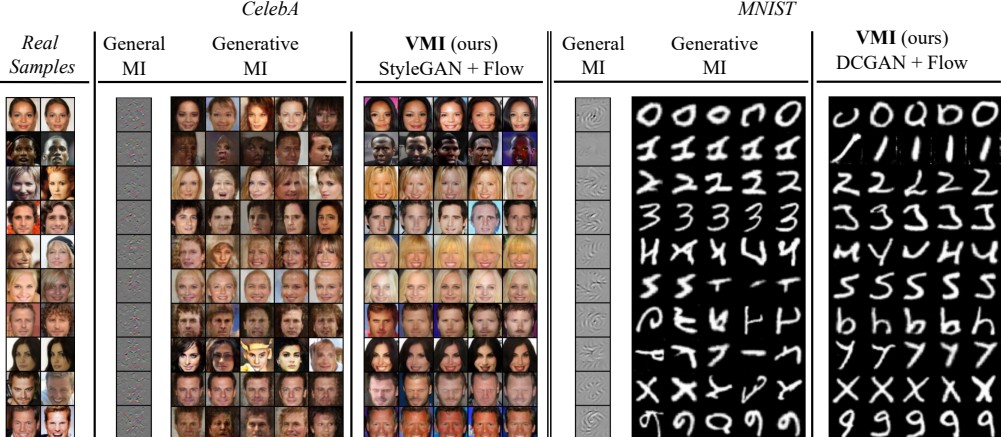

Figure 5: MI attack samples on the first ten identities of CelebA and MNIST. Each row corresponds to a different identity/digit. Qualitative differences are best viewed zoomed in.

| | Generative MI [Zhang et al., 2020] | VMI (ours) | | | | | | | | | |
| | | DCGAN | | | | StyleGAN | | | | | |
| | | $q(\mathbf{z})$ = Gaussian | Flow | | | Gaussian | | | Flow | | |
| | | $\gamma$=0 | 0 | 1e-3 | 1e-1 | 0 | 1e-3 | 1e-1 | 0 | 1e-3 | 1e-1 |
|---|---|---|---|---|---|---|---|---|---|---|---|
| Accuracy | $0.07_{\pm 0.02}$ | $0.24_{\pm 0.05}$ | $0.33_{\pm 0.09}$ | $0.37_{\pm 0.07}$ | $0.13_{\pm 0.03}$ | $0.57_{\pm 0.06}$ | $0.56_{\pm 0.05}$ | $0.23_{\pm 0.03}$ | $\mathbf{0.58}_{\pm 0.06}$ | $0.55_{\pm 0.06}$ | $0.39_{\pm 0.07}$ |
| Precision | $0.51_{\pm 0.04}$ | $0.64_{\pm 0.05}$ | $0.48_{\pm 0.08}$ | $0.52_{\pm 0.06}$ | $0.40_{\pm 0.06}$ | $0.87_{\pm 0.02}$ | $0.88_{\pm 0.02}$ | $0.82_{\pm 0.02}$ | $0.87_{\pm 0.03}$ | $0.87_{\pm 0.03}$ | $\mathbf{0.89}_{\pm 0.06}$ |
| Density | $0.41_{\pm 0.04}$ | $0.67_{\pm 0.08}$ | $0.49_{\pm 0.11}$ | $0.52_{\pm 0.08}$ | $0.38_{\pm 0.06}$ | $1.26_{\pm 0.07}$ | $1.28_{\pm 0.07}$ | $1.14_{\pm 0.06}$ | $1.22_{\pm 0.08}$ | $1.22_{\pm 0.08}$ | $\mathbf{1.31}_{\pm 0.10}$ |
| Recall | $0.21_{\pm 0.04}$ | $0.03_{\pm 0.01}$ | $0.00_{\pm 0.00}$ | $0.01_{\pm 0.01}$ | $0.13_{\pm 0.03}$ | $0.22_{\pm 0.03}$ | $0.25_{\pm 0.03}$ | $\mathbf{0.42}_{\pm 0.03}$ | $0.11_{\pm 0.02}$ | $0.15_{\pm 0.03}$ | $0.21_{\pm 0.04}$ |
| Coverage | $0.83_{\pm 0.03}$ | $0.79_{\pm 0.04}$ | $0.37_{\pm 0.06}$ | $0.67_{\pm 0.06}$ | $0.70_{\pm 0.06}$ | $0.98_{\pm 0.01}$ | $0.98_{\pm 0.01}$ | $\mathbf{0.98}_{\pm 0.01}$ | $0.96_{\pm 0.02}$ | $0.97_{\pm 0.02}$ | $0.98_{\pm 0.02}$ |
| Diversity | $0.52_{\pm 0.05}$ | $0.41_{\pm 0.04}$ | $0.19_{\pm 0.06}$ | $0.34_{\pm 0.06}$ | $0.41_{\pm 0.07}$ | $0.60_{\pm 0.03}$ | $0.61_{\pm 0.03}$ | $\mathbf{0.70}_{\pm 0.03}$ | $0.54_{\pm 0.02}$ | $0.56_{\pm 0.04}$ | $0.59_{\pm 0.05}$ |
| FID | 43.21 | 28.98 | 58.39 | 40.89 | 40.74 | 16.69 | 16.11 | **13.49** | 17.28 | 17.41 | 21.35 |

Table 2: Detailed comparison between attack methods on CelebA.

**Attacking Different Target Classifiers.** After establishing the effectiveness of our VMI attack on different datasets, we conducted experiments to test the effectiveness of our VMI attack against different target classifiers. We further tested two target classifier on CelebA: a VGG network [Simonyan and Zisserman, 2014] trained from scratch on our target dataset, and an adapted ArcFace classifier [Deng et al., 2019], which we denote as ArcFace-NC. The ArcFace is a specialized face identity recognizer pretrained on a large collection of datasets. We adapt it by using the pretrained network as a feature encoder,[3] and adding a nearest cluster classifier in the feature space. This approach of adapting a powerful feature extractor is popular and effective in few-shot classification [Chen et al., 2019; Snell et al., 2017]. The resulting accuracy on held-out test examples in the target dataset were 62.1% (VGG) and 96.1% (ArcFace-NC). Results in Table 3 show that the same observations hold.

**Accuracy vs. Diversity Tradeoff.** Figure 6 shows the accuracy vs. diversity tradeoff in VMI attacks on CelebA. The VMI attacks that used the StyleGAN generally achieved a better accuracy vs. diversity tradeoff than the ones used the DCGAN. We conjecture this is because of the better layer-wise disentanglement that the StyleGAN achieves.

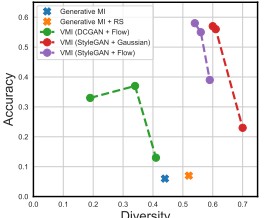

Figure 6: Accuracy and diversity tradeoff on CelebA.

**Effect of Individual Layers.** As motivated in Section 4.1, the extended latent space in Equation 5 allowed our attack to modify each layer which controlled different aspects of an image. Figure 7 shows values for the KL-divergence and entropy for $q_l(\mathbf{z})$ at all layers after VMI optimization in CelebA and ChestX-ray datasets. We can see a general downward trend in KL divergences and an upward trend in entropies of $q(\mathbf{z})$ with respect to layers. This shows that in general the $q(\mathbf{z})$ of the earlier layers were modified more by the optimization than the $q(\mathbf{z})$ of the later layers. This is consistent with our observation in Figure 2 that the earlier layers have more control over features relevant to the identity of a face.

---

[3]Code and model from: https://github.com/TreB1eN/InsightFace_Pytorch

| | VGG | | | | | ArcFace-NC | | | | |
|---|---|---|---|---|---|---|---|---|---|---|
| | | **VMI (ours)** | | | | | **VMI (ours)** | | | |
| | Generative MI | **DCGAN** | | **StyleGAN** | | Generative MI | **DCGAN** | | **StyleGAN** | |
| | [Zhang et al., 2020] | Gaussian | Flow | Gaussian | Flow | [Zhang et al., 2020] | Gaussian | Flow | Gaussian | Flow |
| **Accuracy** | $0.04_{\pm 0.03}$ | $0.16_{\pm 0.09}$ | $0.12_{\pm 0.08}$ | $0.17_{\pm 0.10}$ | $\mathbf{0.33}_{\pm 0.18}$ | $0.21_{\pm 0.08}$ | $0.63_{\pm 0.17}$ | $0.55_{\pm 0.18}$ | $0.52_{\pm 0.13}$ | $\mathbf{0.90}_{\pm 0.07}$ |
| **Precision** | $0.51_{\pm 0.09}$ | $0.60_{\pm 0.10}$ | $\underline{0.74}_{\pm 0.07}$ | $\mathbf{0.80}_{\pm 0.08}$ | $0.79_{\pm 0.11}$ | $0.59_{\pm 0.09}$ | $0.52_{\pm 0.13}$ | $0.53_{\pm 0.14}$ | $\mathbf{0.86}_{\pm 0.04}$ | $0.84_{\pm 0.07}$ |
| **Density** | $0.42_{\pm 0.11}$ | $0.63_{\pm 0.16}$ | $\underline{0.76}_{\pm 0.20}$ | $\mathbf{1.04}_{\pm 0.19}$ | $\underline{1.02}_{\pm 0.24}$ | $0.46_{\pm 0.13}$ | $0.55_{\pm 0.18}$ | $0.64_{\pm 0.21}$ | $\underline{1.26}_{\pm 0.21}$ | $\mathbf{1.26}_{\pm 0.23}$ |
| **Recall** | $0.23_{\pm 0.10}$ | $0.09_{\pm 0.04}$ | $0.02_{\pm 0.02}$ | $\mathbf{0.28}_{\pm 0.06}$ | $0.04_{\pm 0.03}$ | $0.11_{\pm 0.04}$ | $0.01_{\pm 0.01}$ | $0.00_{\pm 0.01}$ | $\mathbf{0.23}_{\pm 0.07}$ | $0.04_{\pm 0.03}$ |
| **Coverage** | $\underline{0.79}_{\pm 0.13}$ | $0.75_{\pm 0.13}$ | $0.77_{\pm 0.12}$ | $\mathbf{0.93}_{\pm 0.08}$ | $0.88_{\pm 0.10}$ | $0.81_{\pm 0.07}$ | $0.69_{\pm 0.11}$ | $0.69_{\pm 0.12}$ | $\mathbf{0.98}_{\pm 0.03}$ | $0.96_{\pm 0.05}$ |
| **Diversity** | $\underline{0.51}_{\pm 0.16}$ | $0.42_{\pm 0.13}$ | $0.39_{\pm 0.12}$ | $\mathbf{0.61}_{\pm 0.10}$ | $\underline{0.46}_{\pm 0.11}$ | $0.46_{\pm 0.08}$ | $0.35_{\pm 0.12}$ | $0.35_{\pm 0.12}$ | $\mathbf{0.60}_{\pm 0.08}$ | $\underline{0.50}_{\pm 0.05}$ |
| **FID*** | 59.27 | 53.85 | 57.88 | **32.51** | 46.13 | 63.63 | 59.23 | 63.37 | **32.94** | 40.22 |

Table 3: Detailed comparison on CelebA for two other target classifiers: VGG, and ArcFace-NC. FID* reported here was computed over 20 identities instead of 100, so the values can not be directly compared with those in Table 2.

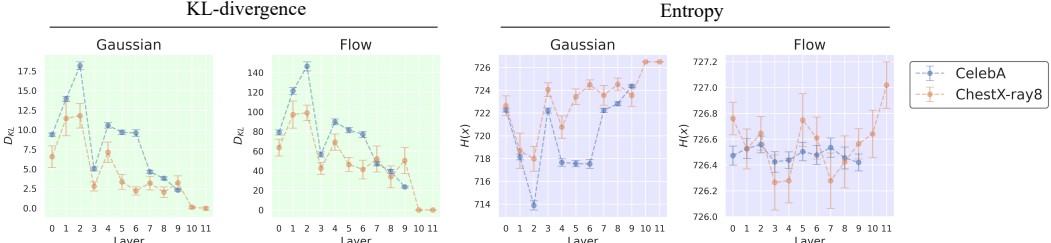

Figure 7: KL and entropy of $q_l(\mathbf{z})$ after VMI attack on CelebA and ChestX-ray at different layers.

# 6 Limitation & Ethical Concerns

While our VMI attack was effective, it relied on the same assumption made in previous works: the attacker had access to a relevant auxiliary dataset [Yang et al., 2019b; Zhang et al., 2020]. A worthwhile future direction is to develop methods that do not depend on this assumption. Methodologically, our proposed VMI attack used a GAN to narrow down the search space of the variational objective, but GANs might not be equally effective for other data modalities such as text, and tabular data. Lastly, our study focused on the white-box setting where the attacker had full access to the model; extensions of the current VMI attack to the grey-box or black-box settings are also worthy of further investigation.

In this work, we focused on a method for improving model inversion attacks, which could have negative societal impacts if it falls into the wrong hands. As discussed in Section 1, malicious users can use attacks like this to break into secured systems and steal information from private data-centers. On the other hand, by pointing out the extent of the vulnerability of current models, we hope to facilitate research that builds better privacy-preserving algorithms, and raise awareness about privacy concerns.

# 7 Conclusion

An effective model inversion attack method must produce a set of samples that accurately captures the distinguishing features and variations of the underlying identity/class. Our proposed VMI attack optimizes a variational inference objective by using a combination of a StyleGAN generator and a deep flow model as its variational distribution. Our framework also provides insights into existing attack methods. Empirically, we found our VMI attacks were effective on three different tasks, and across different target classifiers.

## Acknowledgments and Disclosure of Funding

**Acknowledgement.** We would like to thank reviewers and ACs for constructive feedback and discussion. Resources used in preparing this research were provided, in part, by the Province of Ontario, the Government of Canada through CIFAR, and companies sponsoring the Vector Institute www.vectorinstitute.ai/#partners.

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
