# Appendix A  Proofs

**Proposition 1.** *We have* $D_{\mathrm{KL}}\big(q(\mathbf{z})||p_{\mathrm{TAR}}(\mathbf{z})\big) \geq D_{\mathrm{KL}}\big(\mathbb{E}_{q(\mathbf{z})}[p_{\mathrm{G}}(\mathbf{x}|\mathbf{z})]||\mathbb{E}_{p_{\mathrm{TAR}}(\mathbf{z})}[p_{\mathrm{G}}(\mathbf{x}|\mathbf{z})]\big).$

*Proof.*

$$D_{\mathrm{KL}}\big(q(\mathbf{z})||p_{\mathrm{TAR}}(\mathbf{z})\big) = D_{\mathrm{KL}}\big(q(\mathbf{z})p_{\mathrm{G}}(\mathbf{x}|\mathbf{z})||p_{\mathrm{TAR}}(\mathbf{z})p_{\mathrm{G}}(\mathbf{x}|\mathbf{z})\big) \tag{8}$$

$$\overset{(1)}{\geq} D_{\mathrm{KL}}\big(\mathbb{E}_{q(\mathbf{z})}[p_{\mathrm{G}}(\mathbf{x}|\mathbf{z})]||\mathbb{E}_{p_{\mathrm{TAR}}(\mathbf{z})}[p_{\mathrm{G}}(\mathbf{x}|\mathbf{z})]\big),$$

where $(1)$ uses the fact that for any two arbitrary joint distributions $p(\mathbf{x}, \mathbf{z})$ and $q(\mathbf{x}, \mathbf{z})$, we have

$$D_{\mathrm{KL}}(q(\mathbf{x}, \mathbf{z})||p(\mathbf{x}, \mathbf{z})) = \mathbb{E}_{q(\mathbf{x})}[D_{\mathrm{KL}}(q(\mathbf{z}|\mathbf{x})||p(\mathbf{z}|\mathbf{x}))] + D_{\mathrm{KL}}(q(\mathbf{x})||p(\mathbf{x}))$$

$$\geq D_{\mathrm{KL}}(q(\mathbf{x})||p(\mathbf{x})).$$

$\square$

**Proposition 2.** *The power posterior* $q_\gamma^*(\mathbf{z}) \propto p_{\mathrm{AUX}}(\mathbf{z})\overline{p}_{\mathrm{TAR}}^{\frac{1}{\gamma}}(y|G(\mathbf{z}))$ *is the solution of the following optimization problem:*

$$q_\gamma^*(\mathbf{z}) = \underset{q(\mathbf{z})}{\arg\min}\, \mathcal{L}_{\mathrm{VMI}}^\gamma(q), \tag{9}$$

$$\mathcal{L}_{\mathrm{VMI}}^\gamma(q) := \mathbb{E}_{\mathbf{z}\sim q(\mathbf{z})}[-\log\overline{p}_{\mathrm{TAR}}(y|G(\mathbf{z}))] + \gamma D_{\mathrm{KL}}(q(\mathbf{z})||p_{\mathrm{AUX}}(\mathbf{z})). \tag{10}$$

*Proof.* Suppose $\mathcal{Z}_\gamma$ is the partition function of $q_\gamma^*(\mathbf{z}) = \frac{1}{\mathcal{Z}_\gamma}p_{\mathrm{AUX}}(\mathbf{z})\overline{p}_{\mathrm{TAR}}^{\frac{1}{\gamma}}(y|G(\mathbf{z}))$. The proof follows from the following equality and the fact that $\mathcal{Z}_\gamma$ is independent of $q(\mathbf{z})$.

$$D_{\mathrm{KL}}(q(\mathbf{z})||\frac{1}{\mathcal{Z}_\gamma}p_{\mathrm{AUX}}(\mathbf{z})\overline{p}_{\mathrm{TAR}}^{\frac{1}{\gamma}}(y|G(\mathbf{z}))) = \frac{1}{\gamma}\mathbb{E}_{q(\mathbf{z})}[-\log\overline{p}_{\mathrm{TAR}}(y|\mathbf{x})] + D_{\mathrm{KL}}(q(\mathbf{z})||p_{\mathrm{AUX}}(\mathbf{z})) + \log(\mathcal{Z}_\gamma).$$

$\square$

# Appendix B  Experimental Details

All experiments are run on Nvidia GPUs. The exact softwares can be found in the supplemental code.

## B.1  Datasets

For the MNIST task. The 'letter' split of the EMNIST dataset was used as the auxiliary dataset. The images are resized to are 32x32. For the CelebA task, we split the full Celeb-A dataset into 2 sets:

- a private/target set that contains the most frequent 1000 identities, and
- a public/auxiliary set consisting of the rest 9177 = 10,177 - 1000 identities.

We take the 128x128 center crop of the original images, and resized them to 64x64. For the private dataset, 5 examples were used as unseen test examples to evaluate the classifier accuracy. For the ChestX-ray8 task, the 8 diseased used in the original study [Wang et al., 2017] were used as the target dataset. Here are short descriptions for each of the diseases:

1. Atelectasis: "partial collapse of lung(s)"
2. Cardiomegaly: "enlarged heart"
3. Effusion: "accumulation of fluids 'around' the lungs"
4. Infiltration: "accumulation of fluids 'in' the lungs"
5. Mass: "extra soft tissue"
6. Nodule: "small round mass"
7. Pneumonia: "infection/inflammation that fills the lungs with fluids or pus"
8. Pneumothorax: "complete collapse of lung(s)"

A majority of images in the auxiliary set are from the "normal/healthy" population. In order to preserve the details of the original images, images in this task are resized to 128x128.

## B.2 Target Classifiers

For all the target classifiers, grid search over hyperparameters were done to maximize their accuracy on a validation set. Below we provide the details for the selected hyperparamters used for the MI attack experiments.

**MNIST.** The target classifier for CelebA was a ResNet10. It was trained using Adadelta (learning rate=1e-1, batch size=32) for 13 epochs. Learning rate decayed by a factor of 0.7 at every epoch. The best validation accuracy for the 10-way classification problem was 98.1%.

**CelebA.** The target classifier for CelebA was a ResNet34. It was trained using SGD with Nestrov momentum (learning rate=1e-1, batch size=64, momentum=0.9, weight decay=5e-4) for 200 epochs. Learning rate decayed by a factor of 0.2 at 60, 120 and 160 epochs. CutOut [DeVries and Taylor, 2017] was used as data augmentation. The best validation accuracy for the 1000-way classification problem was 69.0%.

**Chest-Xray-8.** The target classifier for CelebA was a ResNet34. It was trained using SGD with Nestrov momentum (learning rate=1e-1, batch size=64, momentum=0.9, weight decay=5e-4) for 200 epochs. Learning rate decayed by a factor of 0.2 at 60, 120 and 160 epochs. Translation was used as data augmentation. The best validation accuracy for the 8-way classification problem was 45.3%.

## B.3 Evaluation Classifiers

**MNIST.** For MNIST, the evaluation classifier had the same model structure and hyperparameters as the target classifier, but was trained with a different random seed.

**CelebA.** For CelebA, we started with a pretrained checkpoint from a large scale facial recognition task [4], and further finetuned it on our private training set after replacing the final classification layer with a randomly initialized linear layer. The final accuracy of our evaluation classifier on the unseen set was 97%.

**ChestX-ray.** For ChestX-ray, we followed the recommendation from the original paper [Wang et al., 2017] and started with a ResNet50 pretrained on ImageNet, and finetuned on the target dataset. The final accuracy on the unseen test set was 50.3%.

## B.4 Flow Details

In our experiments, we use the Glow model from Kingma and Dhariwal [2018] as our variational distribution $q(z)$. As discussed in Section 4.2, we treat the latent vectors as 1x1 images, and remove the squeezing layers that were designed to reduce image sizes. The other hyperparameters can be found in the following table:

| Hyperparameter | Value |
|---|---|
| Flow Permutation | Random Shuffle |
| Flow Coupling Type | Additive |
| # of Total Invertible Blocks | 30 |
| # of Conv Layers per Block | 3 |
| # of Channels per Conv Layer | 100 |
| Activation Function | ELU |

# Appendix C   Additional Results

Detailed results including all metrics for MNIST and ChestX-ray are shown in Table 4, and Table 5 respectively. The attack samples for ChestX-ray are in Figure 8.

---

[4]the IR-SE50 checkpoint from `https://github.com/TreB1eN/InsightFace_Pytorch`.

| | General MI [Hidano et al., 2017] | Generative MI [Zhang et al., 2020] | VMI (ours) DCGAN | |
|---|---|---|---|---|
| | | | Gaussian | Flow |
| **Accuracy** | $0.00 \pm 0.00$ | $0.92 \pm 0.02$ | $0.93 \pm 0.06$ | $\mathbf{0.95} \pm 0.02$ |
| **Precision** | $0.00 \pm 0.00$ | $0.25 \pm 0.14$ | $0.26 \pm 0.13$ | $\mathbf{0.35} \pm 0.15$ |
| **Density** | $0.00 \pm 0.00$ | $0.09 \pm 0.07$ | $0.11 \pm 0.06$ | $\mathbf{0.14} \pm 0.09$ |
| **Recall** | $0.00 \pm 0.00$ | $0.39 \pm 0.12$ | $\mathbf{0.54} \pm 0.12$ | $0.25 \pm 0.10$ |
| **Coverage** | $0.00 \pm 0.00$ | $0.20 \pm 0.12$ | $0.17 \pm 0.08$ | $\mathbf{0.24} \pm 0.12$ |
| **Diversity** | $0.00 \pm 0.00$ | $0.29 \pm 0.17$ | $\mathbf{0.36} \pm 0.15$ | $0.24 \pm 0.16$ |
| **FID** | 376.7 | 88.91 | 82.52 | **77.73** |

Table 4: MNIST: comparing baseline and our attacks.

| | General MI [Hidano et al., 2017] | Generative MI [Zhang et al., 2020] | VMI (ours) DCGAN | | StyleGAN | |
|---|---|---|---|---|---|---|
| | | | Gaussian | Flow | Gaussian | Flow |
| **Accuracy** | $0.23 \pm 0.29$ | $0.28 \pm 0.24$ | $0.36 \pm 0.25$ | $0.42 \pm 0.28$ | $0.54 \pm 0.24$ | $\mathbf{0.69} \pm 0.23$ |
| **Precision** | $0.00 \pm 0.00$ | $0.15 \pm 0.09$ | $0.20 \pm 0.05$ | $0.08 \pm 0.13$ | $\mathbf{0.30} \pm 0.09$ | $0.15 \pm 0.12$ |
| **Density** | $0.00 \pm 0.00$ | $0.06 \pm 0.03$ | $0.08 \pm 0.03$ | $0.02 \pm 0.03$ | $\mathbf{0.18} \pm 0.06$ | $0.08 \pm 0.06$ |
| **Recall** | $0.00 \pm 0.00$ | $0.04 \pm 0.04$ | $0.07 \pm 0.06$ | $0.00 \pm 0.00$ | $\mathbf{0.32} \pm 0.10$ | $0.05 \pm 0.04$ |
| **Coverage** | $0.00 \pm 0.00$ | $0.14 \pm 0.07$ | $0.17 \pm 0.04$ | $0.00 \pm 0.01$ | $\mathbf{0.43} \pm 0.09$ | $0.12 \pm 0.08$ |
| **Diversity** | $0.00 \pm 0.00$ | $0.09 \pm 0.08$ | $0.12 \pm 0.07$ | $0.00 \pm 0.01$ | $\mathbf{0.38} \pm 0.13$ | $0.09 \pm 0.09$ |
| **FID** | 499.54 | 142.66 | 104.23 | 265.14 | **63.78** | 123.17 |

Table 5: ChestX-ray8: comparing baseline and our attacks.

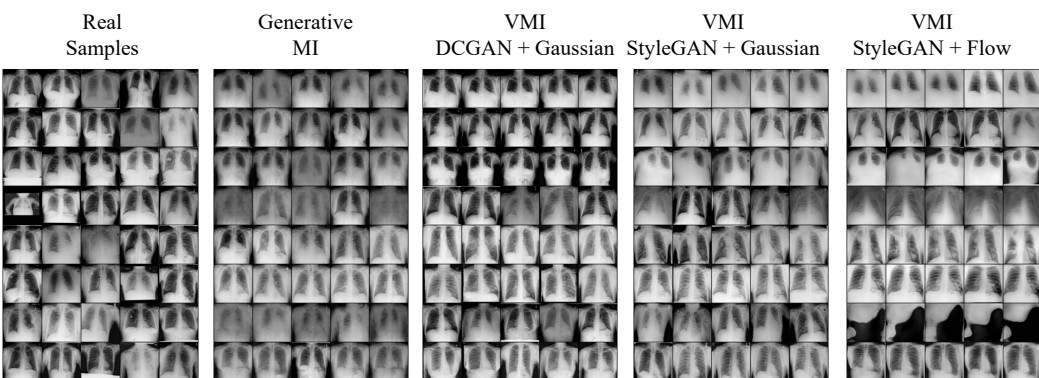

| Real Samples | Generative MI | VMI DCGAN + Gaussian | VMI StyleGAN + Gaussian | VMI StyleGAN + Flow |

Figure 8: MI attack samples on ChestXray. Each row corresponds to a different disease. Best viewed zoomed in.