# OpenReview forum: "Variational Model Inversion Attacks"
_NeurIPS.cc/2021/Conference — NeurIPS 2021 Poster_

### Official Review · Reviewer_Zyff · 2021-07-11

**Rating:** 7
**Confidence:** 3

**Summary:**

This paper frames model inversion attacks as a variational inference problem, leading to the derivation shown in Equation one. Following prior work, they also assume access to a common generator, than can be learned on auxiliary data and can capture a shared conditional distribution $p_G(x|z)$. With this assumption, they can derive a practical objective as shown in Equation 3, which maximizes (in code space) the attack accuracy (first term) and the realism / diversity of the generated samples (KL term).

From an empirical standpoint, they experiment using both DGGAN and StyleGAN as their pretrained generator, and using either a gaussian or flow model to form their $q(z)$. They show that their model outperforms prior work across MNIST, CelebA and CXR datasets in terms of attack accuracy, FID and diversity.


**Limitations And Societal Impact:**

Yes, the authors noted their primary limitation (assumption 1) and indicated multiple useful areas for future work. They also noted the possible ethical downsides of this line of work (people doing more powerful attacks).

**Main Review:**

Originality:
This study offers a well-motivated framing of model inversion attacks (i.e as variational inference), and this framing offers a useful lense to unify the prior work and leads to a natural objective. This offers a novel perspective to this line of work.

Quality:
The core claims of this paper (i.e improving the performance of model inversion attacks) are well supported by their empirical results across MNIST, CelebA and CXR.

Clarity:
As a whole, the paper is well written and easy to follow. There were a few areas that I thought could benefit from improved exposition:

Section 4.2:
      There really isn’t enough detail here to get a concrete idea of what is being used without just looking through the code. If there isn’t enough space, this should maybe be detailed better in an appendix.

Section 4.1:
      This description was difficult to follow.


Significance:
This paper offers a useful new perspective on model inversion attacks, and their constriction leads to improved attacks. I think this study is useful for this ongoing work in better understanding, developing and defending against model inversion attacks.


**Time Spent Reviewing:**

2

---

> ### Author Response · Authors · 2021-08-10
> **response**
>
> We thank the reviewer for the review.
> The reviewer is concerned with a few places of inclarity in the writing.
> We address them here, and intend to add the clarification into the updated paper and appendix.
>
> **Q:** *“Section 4.2: There really isn’t enough detail here to get a concrete idea of what is being used without just looking through the code. If there isn’t enough space, this should maybe be detailed better in an appendix.”*
> **A:** We will add details for the flow model used in our experiment in the Appendix.  Here is what we plan to add:
> “
> In our experiments, we use the Glow model from Kingma & Dhariwal as our variational distribution $q(z)$.  As discussed in Section 4.2, we treat the latent vectors as 1x1 images, and remove the squeezing layers that were designed to reduce image sizes.  The other hyperparameters can be found in the following Table:
>
> | Hyperparameter               | Value          |
> |------------------------------|----------------|
> | Flow Permutation             | Random Shuffle |
> | Flow Coupling Type           | Additive       |
> | # of Total Invertible Blocks | 30             |
> | # of Conv Layers per Block   | 3              |
> | # of Channels per Conv Layer | 100            |
> | Activation Function          | ELU            |
>
>
> [1] Kingma, Diederik P., and Prafulla Dhariwal. "Glow: Generative flow with invertible 1x1 convolutions." arXiv preprint arXiv:1807.03039 (2018).
> “
>
> Again, we thank the reviewer for this comment. We also agree that adding these details in the paper helps the reader.
>
> **Q:** *“Section 4.1: This description was difficult to follow”*
> **A:** We will add the following paragraph as the leading paragraph to Section 4.1:
> “
> In the previous section, we derived a practical MI objective from the perspective of variational inference.  Similar to previous works [1, 2], VMI requires a common generator that is trained on a relevant auxiliary dataset.  In this section, we first provide a quick description of the most common choice, DCGAN. Then, we introduce StyleGAN, which has an architecture that allows for fine-grained control.  Finally, we describe how to adapt our VMI objective to leverage this architecture by using a separate approximate distribution at each layer of StyleGAN’s synthesis network.
>
> [1] Zhang, Yuheng, et al. "The secret revealer: Generative model-inversion attacks against deep neural networks." Proceedings of the IEEE/CVF Conference on Computer Vision and Pattern Recognition. 2020.
> [2] Si Chen, Ruoxi Jia, and Guo-Jun Qi. Improved techniques for model inversion attack. arXiv preprint arXiv:2010.04092, 2020.
> ”
>
> We hope our response has addressed the reviewer’s concern.

---

### Official Review · Reviewer_He1L · 2021-07-15

**Rating:** 6
**Confidence:** 4

**Summary:**

The paper at hand propose a new model inversion attack that recovers the training dataset that was used to learn a supervised model. The proposed attack uses generative adversarial networks (namely a DCGAN and a StyleGAN) and variational inference to invert the probability distribution $p(y|x)$ that is learned by the classifier. The authors also describe how the proposed VMI framework can be seen as an unified framework of existing model inversion attacks. Experimental evaluation was performed on MNIST, the ChestX-ray dataset and the CelebA dataset.

**Limitations And Societal Impact:**

Yes.

**Main Review:**

REASONS FOR SCORE:

- The reported experimental results look promising, however I have some concerns about the evaluation:
    1. As the generator is in some cases also trained on the actual input distribution it would be important to also compare the reported FID scores to FID scores of the generator before the attack was performed.
    2. Related works (e.g., [2] which is also cited by the authors) also evaluate on the CIFAR10 dataset. The authors might consider extending their experiments to also include that dataset.
    3. What is the test accuracy of the target classifiers before the attack was performed?
    4. How does VMI compare to General MI in terms of computational runtime?

- The structure of the paper is reasonable, but the mathematical writing can be improved at many places:
    1. The notation $G : z \rightarrow x$ (used, e.g., in l. 129 and l. 135) is not correct, as $x$ and $z$ are random variables and not sets.
    2. In l. 119: What is p? (I suppose $p^\text{tar}$ is meant.)
    3. In l. 143f: "$\mathcal{J}(q)$ can be rewritten as..." (I think "rewritten" is the wrong word here, because the following inequality is a lower bound!)
    4. In Eqn. 5: The authors should write $x^*$ instead of $x'$ (which would be more consistent with the notation of the rest of the paper)
    5. In l. 136: It would be better to write $p^G$ instead of $p_G$ (as it would be more consistent with $p^\text{tar}$, $p^\text{aux}$).

- Zhang et al. [1] are able to use auxiliary information about the retrieved images, e.g., blurred versions of the images or partly occluded images. How can auxiliary information be incorporated into this framework?

CONCLUSION:

Overall, I lean towards rejection. The proposed method sure has its merits, but the writing of the paper can be improved and the discussion of the performed experiments lacks some details.

MINOR REMARKS:

- Subsections not always in title-case (for example: "Relationships with existing methods" vs. "Model Class of q(z)")
- l.219: \gamma=0 not in math mode

REFERENCES:

[1] Zhang, Yuheng, et al. "The secret revealer: Generative model-inversion attacks against deep neural networks." Proceedings of the IEEE/CVF Conference on Computer Vision and Pattern Recognition. 2020.

[2] Si Chen, Ruoxi Jia, and Guo-Jun Qi. Improved techniques for model inversion attack. arXiv preprint arXiv:2010.04092, 2020.


**Time Spent Reviewing:**

4

---

> ### Author Response · Authors · 2021-08-10
> **response**
>
> We thank the reviewer for the thoughtful review.  We address each of the concerns below.
>
> **Q:** *“FID of generator before using VMI”*
> **A:** Thanks for the suggestion. We will add this information in the updated paper.
> FID, as discussed in the beginning of Section 5 is a metric for overall sample quality, but not an attack-specific metric.
> The FIDs of samples from the generator without applying any MI attack techniques are listed in the following table:
>
> |          | MNIST | CelebA |  CXR |
> |----------|:-----:|:------:|:----:|
> | DCGAN    |  57.4 |  31.3  | 96.3 |
> | StyleGAN |   -   |  19.2  | 57.0 |
>
> These FIDs are measured between samples from the pretrained generator and the target dataset, and directly comparable to other FIDs reported in the paper (e.g. Table 1).  For StyleGAN on both (CelebA, and CXR), the FIDs here are worse (higher) than in Table, suggesting that VMI not only did not sacrifice overall sample quality, but actually improved upon it. The same held for DCGAN on the CXR benchmark.  This was not the case for the DCGAN variant on MNIST and CelebA, which suggests room for improvement of the overall sample quality.  However, it is still worth noting that this baseline isn’t an attack baseline since it cannot perform targeted attacks, but rather informs us which methods can better preserve (or improve) overall sample quality.  In this case, our proposed VMI in the augmented latent space of StyleGAN led to improvements.
>
> **Q:** *”Related works also evaluate on the CIFAR10 dataset. The authors might consider extending their experiments to also include that dataset”*
> **A:** We did not include results for CIFAR10 for the following reasons.  We experimented with 3 different datasets. A toy dataset on hand-written characters, and two datasets with real world motivations from the perspective of privacy attacks.  We believe there are reasons that make CIFAR10 an inappropriate benchmark. First, it does not have any motivation from the perspective of privacy attacks. Second, it is also not suitable as a toy task. There are only a total of 10 classes in CIFAR10, and the latent components across the classes are not necessarily shared (e.g. ‘frogs’ and ‘cars’). Unlike in handwritten characters, the strokes like lines and curves are shared among the classes.  In sum, we find it to be unsuitable for benchmarking the MI attack problem, and don’t see clear value in including CIFAR10 as a 4th dataset. If the reviewer has other reasons for running the CIFAR10 experiment, we are happy to run it and provide the results during the discussion period.
>
> **Q:** *“What is the test accuracy of the target classifiers before the attack was performed?”*
> **A:** We have discussed this in Appendix B.2.
>
> **Q:** *“How does VMI compare to General MI in terms of computational runtime?”*
> **A:**  Previous optimization based methods are costly in the sense that every time they need to perform the attack and generate new samples, they have to run gradient descent again.  In the case of VMI, it is costly when we first learn our variational distribution, which also requires running gradient descent.  Once it’s learned, performing the attack and generating new samples are as simple as a single forward pass.  As an example, during the development of our project, VMI benefited us in that once a checkpoint for $q(z)$ is stored, evaluation is very fast, whereas evaluating baselines was always costly.
>
> **Q:** *“Comments on mathematical writing”*
> **A:** First, we’d like to thank the reviewer again for making these suggestions.  Here are our responses:
> 1. Thanks for catching this mistake. We will fix it.
> 2. In line 119 we simply describe the generic KL-divergence.  The exact target distribution is made clear in Equation 1, which is p^{tar}(\mathbf{x}|y)
> 3. We will rephrase it to "an upper bound of J(q) can be derived:"
> 4. & 5. we will incorporate them in our revision.
>
> **Q:** *“How can auxiliary information be incorporated into this framework?”*
> **A:** We see this as an interesting extension to our project, because it is orthogonal to our contributions.  Just like in Zhang et al., they basically had different architectures for different cases (i.e. unconditional vs conditional).  One can simply use a conditional generator into the VMI framework.
>
> **Q:** *“discussion of the performed experiments lacks some details.”*
> **A:** We will include the details of the hyper-parameters used in our experiments in an Appendix section. We will also open-source our code as well as the re-implementation of the baselines used in our paper.
>
> We hope that our response addressed the reviewer’s concerns.

---

> > ### Comment · Reviewer_He1L · 2021-08-26
> > **Answer to rebuttal**
> >
> > I would like to thank the authors for their follow-up. The new experimental results are well appreciated.
> >
> > Re: "FID of generator before using VMI"
> >
> > I got some concerns about the new FID values of the generative before the attack was performed. In Table 1 in the paper the following values are stated:
> >
> > |                         | *MNIST*|*CelebA*|*CXR*|
> > |-------------------------|--------|--------|------|
> > |VMI w/ DCGAN (Eqn 3)     |   77.73| 40.89  |265.14|
> > |VMI w/ StyleGAN (Eqn 4)  |   -    | 17.41. |123.17|
> >
> > When being compared with the new provided FID scores one can see that the attack only improves the score for the DCGAN on CelebA and for the StyleGAN on CelebA. All other vales are significant higher than before the attack. Thus, I do not agree to your conclusion that VMI did not sacrifice overall sample quality.
> >
> > However, I will raise my score to 6 as most of the other points are appropriately addressed in the rebuttal.

---

> > > ### Author Response · Authors · 2021-08-27
> > > **response**
> > >
> > > Dear reviewer,
> > >
> > > Thank you for your response and for raising your score. We are happy that most of your concerns have been addressed.
> > > Also thanks for catching our mistake in misreading the FID values of the CXR dataset in the Table of our rebuttal. The VMI FID is indeed worse than the generator FID in all cases except StyleGAN CelebA.
> > >
> > > To summarize, while the proposed VMI attack achieved better FIDs than the baseline attacks, the FIDs are usually worse compared to the generator FIDs.  This is expected, as the VMI trades off image quality and diversity in order to generate images that are closer to the target distribution; the generated images achieve higher accuracy as measured by the target classifier. This suggests that the overall sample quality can be improved.  As discussed in our Limitation section, methods that can better account for the distribution shifts between the auxiliary and target dataset can lead to further improvements. We will include these additional experiments and discussions in the final paper.
> > >
> > > Thanks again,
> > > Authors

---

### Official Review · Reviewer_uEuV · 2021-07-20

**Rating:** 5
**Confidence:** 4

**Summary:**

This paper frames the model inversion problem as a variational inference problem, where the goal is to approximate the posterior distribution p^{tar}(x|y) from the known target classifier p^{tar}(y|x). From a class of candidate distributions q(x), the aim is to find one that is close to p^{tar}(x|y), and this is achieved by minimizing the KL-divergence between the 2 distributions. Such a formulation allows the authors to make a principled derivation of an objective function that encourages both the optimized distribution to produce images likely to be classified correctly by the target classifier, and the produced images to be realistic. It is also a unified framework for existing methods of generative model inversion.

Additionally, the authors propose using the StyleGAN architecture as a pre-trained generator for the model inversion task. This allows intermediate layers (as opposed to just the input latent code in previous formulations) of the GAN to be optimized during the search for q*, improving the expressiveness of the GAN. The StyleGAN architecture also means that the attack can focus on layers whose representations control the semantics of the final image.


**Limitations And Societal Impact:**

yes, this is discussed in sec 6.



**Main Review:**

The techniques proposed in this paper are not entirely novel by themselves (e.g. distributional recovery in model inversion has been proposed in Si Chen, Ruoxi Jia, and Guo-Jun Qi. Improved techniques for model inversion attack.) However, the novelty is the theoretical framework used to motivate such techniques - framing the model inversion problem as approximating the posterior p^{tar}(x|y) distribution.

The proposal to use StyleGAN as the generator architecture makes sense, since individual layers can potentially control different features related to identity. However, this is not particularly groundbreaking. One can imagine trying out other architectures used in unsupervised disentangled representation learning as well (e.g. InfoGAN).

In the derivation of the objective function, a number of assumptions are made. Assumption 1 assumes that a generator trained on the auxiliary dataset can capture the distribution of the target data. However, given that the target data and the auxiliary data are meant to be disjoint (should not contain the same classes), it is not immediately obvious that this assumption holds. For example, the generator trained on the ‘letters’ split of the EMNIST dataset is unlikely to capture the distribution of the ‘digits’ images in the MNIST dataset.

There are also some concerns about the reported results: in table 2, Generative MI on CelebA dataset is reported to have an accuracy of only 6%. However, in the original paper cited (Zhang 2020), the authors reported an accuracy of 44%. What is the reason for the discrepancy? If Generative MI can achieve 44% accuracy, then VMI w/DCGAN which achieves 37% accuracy is not better. The same applies to the Chest X-ray dataset which in the Zhang et al has an accuracy of 71% but is reported to be 28% in this work. Though there are some differences in training data between the implementations, it does not explain such a huge discrepancy in results.

========== after rebuttal ===========



There is large discrepancy between the results reported in this paper and the original GMI paper Zhang et al. 2020.


Given such a large discrepancy (6% reported in this submission vs 44% reported in Zhang et al. 2020), and VMI w/ DCGAN actually has poorer accuracy than the original GMI (37% vs 44%), it would be difficult for me to further increase my rating.







**Time Spent Reviewing:**

6 hours

---

> ### Author Response · Authors · 2021-08-10
> **response**
>
> We thank the reviewer for the thoughtful review. Please read our responses below.
>
> **Q:** *“The techniques proposed in this paper are not entirely novel by themselves (e.g. distributional recovery in model inversion has been proposed in Si Chen, Ruoxi Jia, and Guo-Jun Qi. Improved techniques for model inversion attack.)”*
> **A:** We believe that our contributions are fundamentally different from that of Chen et al. Our method, VMI, is derived from the variational inference stand-point, and provides the full derivation which was absent in previous works. VMI also provides a unified framework that encompasses these previous methods (see Section 4.3).  Furthermore, Chen et al. only considered using a Gaussian latent distribution, whereas VMI explored using a flow as the latent distribution. In addition, we also proposed performing variational inference in the augmented latent space of a StyleGAN, which has not been explored before. Empirically, we showed that our results consistently outperform Chen et al. (i.e., in our ablations, using VMI using StyleGAN and flow was always better than the variant using DCGAN and Gaussian) in terms of accuracy across a range of datasets and target classifiers.
>
> **Q:** *“One can imagine trying out other architectures used in unsupervised disentangled representation learning as well (e.g. InfoGAN).”*
> **A:** Thanks for the suggestion.  We agree that using other types of base generators is an interesting direction.  As we discussed in the paper, we chose the StyleGAN architecture specifically because of the layerwise decoder.  Also, VMI optimizes in the extended latent space, which allows us to achieve more fine-grained control.  This would not be achieved if we used a pretrained generator like InfoGAN, which does not have an architecture that produces more disentangled latent codes at different layers.
>
> **Q:**  *”it is not immediately obvious that this assumption holds. For example, the generator trained on the ‘letters’ split of the EMNIST dataset is unlikely to capture the distribution of the ‘digits’ images in the MNIST dataset.”*
> **A:**  Intuitively, if we believe that the observations are composed of the same underlying building blocks, such as eyes and noses in the case of faces, and straight lines and curvy strokes in the cases of written characters, then the assumption is reasonable.
> Please note that we only explicitly state an assumption that was implicitly made in prior works.  The same intuitive idea of using a common generator based on this same assumption was used in previous works including Zhang et al, and Chen et al.  So, in fact, our method is not any less applicable than the previous methods due to this assumption.
> Lastly, as we stated in our Limitation section, it is true that if the auxiliary dataset is irrelevant to the target dataset, then this assumption breaks and hence VMI would not work. The other existing methods using a pretrained generator would not work for the same reason. This concerns the more general problem of coping with distribution shifts, which we believe is an important topic for future work.
>
> **Q:** *“Though there are some differences in training data between the implementations, it does not explain such a huge discrepancy in results.”*
> **A:** First, the code of the previous methods (Zhang et al., and Chen et al.) are not publicly available. We spent a significant amount of time trying to reproduce the baselines.  We also reached out to the authors to ask for reproducing details and code. Although they responded, they did not provide the details nor the code. In the end, we reported the best results we were able to achieve using their methods as described in the papers, and even included different variants (i.e., with and without rejection sampling, which is an extra step that our method does not depend on).
> In the problem of model inversion attack, the evaluation number is dependent on the exact target classifier under attack, and the data split (i.e., which classes are in the auxiliary set, and which are in the private set).  On CelebA, using VMI with StyleGAN and Flow, the three target classifiers we considered resulted in accuracies of 33% for the VGG classifier,  90% for the ArcFace-NC classifier (in Table 3), and 55% for the ResNet classifier (in Table 1).  So the difference in target accuracy caused by the target classifier can be substantial.  Thus, numbers are not directly comparable when these settings change, which is the case when directly comparing numbers in our paper to theirs.
> Our response to this concern is summarized by 3 points: 1. there is discrepancy between our reported baseline number, and the original number; however this can be caused by the exact implementation of the problem, 2. to solidify the improvement brought about by VMI, we experimented with a number of target classifiers, and on different datasets, and 3. the best way to facilitate future research is to release code, and we plan to make our code including our method and the baselines public upon publication. Also, we will point out the discrepancy in the results in the updated paper.
>
> We appreciate the constructive feedback in this review and hope that our response has addressed all the concerns of the reviewer.

---

> > ### Comment · Reviewer_uEuV · 2021-09-02
> > **RE: response**
> >
> > Thank you authors for the effort to respond to my concerns.
> >
> > Regarding the large discrepancy between the results reported in this paper and the original GMI paper Zhang et al. 2020, we indeed have tried to replicate the results in Zhang et al. 2020, and we are able to obtain similar results as in Zhang et al. 2020 for CelebA  (Target: ResNet). We have followed the appendix in Zhang et al. 2020 for the GAN architecture.
> >
> > While it is true that it indeed required some effort and clarification from the authors of Zhang et al. 2020 in order to replicate the results, given the large discrepancy (6% reported in this submission vs 44% reported in Zhang et al. 2020), and VMI w/ DCGAN actually has poorer accuracy than the original GMI (37% vs 44%), it would be difficult for me to further increase my rating.

---

> > > ### Author Response · Authors · 2021-09-02
> > > **Response to the reviewer's comment**
> > >
> > > Thank you for your feedback. It seems that the only remaining concern is with respect to our efforts in reproducing the results of Zhang et al. 2020.
> > >
> > > As we described in our original response, the discrepancy can come from many sources including the dataset split (i.e., which classes were in the private dataset) and the exact classifier used. We would like to again point out that all our attempts in obtaining the experimental details of Zhang et al. 2020 by either contacting the authors or finding online repositories were not successful. Without having the same dataset, classifier, and other experimental details, the comparison of accuracies is essentially an apple to orange comparison. In the final paper, we will include a prominent paragraph that includes the original results of  Zhang et al. 2020, and summarizes our rebuttal discussion about the discrepancy between the results.  We will also release our code and dataset to facilitate better reproducibility for future research in this area.

---

### Decision · Program_Chairs · 2021-09-27

**Decision:**

Accept (Poster)

**Comment:**

The paper proposes a well-motivated method with technical novelty, and good performance is shown.
Reviewer's major concerns, including the questionable assumption and the seemingly inconsistent baseline results with the original papers, have been addressed.
Mathematical notation should be improved in the camera-ready version.